# DYNATUNE: DYNAMIC TENSOR PROGRAM OPTIMIZATION IN DEEP NEURAL NETWORK COMPILATION

**Minjia Zhang**,[*] **Menghao Li\*, Chi Wang & Mingqin Li**
Microsoft Corporation
`{minjiaz,t-meli,wang.chi,mingqli}@microsoft.com`

## ABSTRACT

Recently, the DL compiler, together with *Learning to Compile* has proven to be a powerful technique for optimizing deep learning models. However, existing methods focus on accelerating the convergence speed of the individual tensor operator rather than the convergence speed of the entire model, which results in long optimization time to obtain a desired latency. In this paper, we present a new method called DynaTune, which provides significantly faster convergence speed to optimize a DNN model. In particular, we consider a Multi-Armed Bandit (MAB) model for the tensor program optimization problem. We use UCB to handle the decision-making of time-slot-based optimization, and we devise a Bayesian belief model that allows predicting the potential performance gain of each operator with uncertainty quantification, which guides the optimization process. We evaluate and compare DynaTune with the state-of-the-art DL compiler. The experiment results show that DynaTune is 1.2–2.4 times faster to achieve the same optimization quality for a range of models across different hardware architectures.

## 1 INTRODUCTION

The enormous computational intensity of Deep Neural Network (DNN) models has attracted great interest in optimizing their performance. Popular deep learning (DL) frameworks such as Py-Torch (Paszke et al., 2019) and TensorFlow (Abadi et al., 2016) adopt custom optimized kernels such as Intel MKL-DNN or Nvidia cuDNN (Chetlur et al., 2014) as back-end. However, given the increasing complexity of tensor operations in DNNs and the volatility of DL algorithms, it calls for developing fast and automated compilation frameworks to handle the unprecedented amount of innovations. To imitate or even exceed the success of hand-optimized libraries, recent research has developed neural network compilers, such as XLA (Leary & Wang, 2017), Glow (Rotem et al., 2018), Tensor Comprehension (Vasilache et al., 2018), and TVM (Chen et al., 2018a). Among them, TVM has shown superior performance improvements using a technique called *Learning to Compile* (AutoTVM) (Chen et al., 2018b). AutoTVM optimizes the code by generating many versions of a tensor operator and chooses the best through a learned cost model and search over a large space of code transformation choices.

While the *Learning to Compile* approach produces highly optimized code of DNN models, it suffers from excessively long optimization time. As an example, although AutoTVM is able to demonstrate close to $2\times$ performance improvement over TensorFlow on ResNet-18, the optimization time can take several hours or even tens of hours (Chen et al., 2018b). The long optimization time hinders the turnaround time and even puts the practical utility of the current compiler-based solutions into question. Recent works strive to reduce the optimization time by improving the search strategy for the code transformation plan and lowering the hardware measurement cost (Ahn et al., 2020; Adams et al., 2019). However, these approaches mostly focus on accelerating the convergence speed of optimization at the individual tensor operator level (e.g., Conv2D, batched GEMM), which do not necessarily solve the issue of slow convergence and long optimization time of the entire model, often containing tens of tensor operators.

Different from existing methods, we introduce DynaTune, a DL code optimization algorithm that minimizes the sum of the execution time of all operators in a model as much as possible and as

---

[*]Both authors contributed equally. Order of appearance is random.

quickly as possible. Specifically, the contributions of our paper consist of (1) a preliminary analysis that reveals the challenges and opportunities from existing DL code optimization strategies, (2) a time-slot-based optimization scheme, which simultaneously explores different operators and learns in an online manner that allows to dynamically switch to optimizing more promising tensors operators. (3) a Bayesian belief model that predicts future performance gains of operators, which helps make better decisions and expedites the convergence speed. (4) a detailed evaluation of the proposed algorithm with modern DNNs (ResNet-18, VGG, SqueezeNet, Transformer) on both CPU and GPU. Compared with the leading framework, AutoTVM, DynaTune is 1.2–2.4× times faster to obtain the same levels of optimization.

## 2 BACKGROUND

**DL compilation pipeline.** A typical DL compiler contains multiple passes to optimize a model trained by popular DL frameworks such as TensorFlow (Abadi et al., 2016), PyTorch (Paszke et al., 2019), or MXNET (Chen et al., 2015), as shown in Fig. 1. In the first pass (box with dotted line), the compiler frontend applies target-independent and white-box target-dependent optimizations that do not include a measure of actual execution time. The target-independent passes perform optimizations such as operator fusion and data layout transformation, and the white-box target-dependent optimizations apply heuristic rules for code transformation based on domain knowledge. Recent work such as AutoTVM (Chen et al., 2018b) extends the pipeline with another pass, a black-box target-dependent pass, which uses learning machinery to perform optimizations.

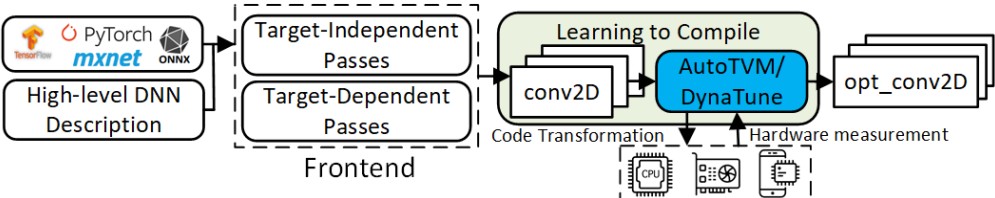

Figure 1: Compilation pipeline.

**Black-box target-dependent pass.** In this pass, the compiler converts code transformation decisions as code templates. A template contains knobs that control various aspects of the optimization (e.g., memory tiling, loop transformations, vectorization) and determines whether the code (1) fully utilizes the internal parallelism within processors, (2) uses the shared memory wisely, and (3) maximizes data locality. Due to the large transformation space, the compiler makes use of an auto-tuner (with an optimization algorithm) and real hardware measurements to find the best transformation on target hardware (e.g., CPU, GPU, ARM, or IoT devices) (Chen et al., 2018b).

## 3 CHALLENGES AND MOTIVATIONS

This section presents several studies that reveal the challenges of existing DL compilation that guided our design in Section 4.

**Challenge 1. Existing DL compilation focuses on accelerating the convergence speed of individual tensor operator instead of the entire model, resulting in slow convergence and long optimization time.** Prior work (Chen et al., 2018a;b; Vasilache et al., 2018; Ahn et al., 2020) optimizes one tensor operator at a time in a predefined order (e.g., in declaration order). However, such an optimization strategy is not always appropriate in practice. For example, there is often an extreme performance difference (e.g., an order of magnitude) between optimized and unoptimized operators. If we optimize operators sequentially, the overall model inference time stays high as long as there are still unoptimized operators. As a result, practitioners may need to wait until all tensor operators have finished optimization to get the desired latency, which results in long optimization time. With the active research that has been pushing the model size to millions or even billion-scale parameters with a training time of only a few hours or less than one hour (Yamazaki et al., 2019; Goyal et al., 2017; You et al., 2017; Lin et al., 2019; Shoeybi et al., 2019; You et al., 2019), it becomes even more prominent to reduce the inference optimization cost of the current solution. Furthermore, since major players in the industry have adopted many of these DL compilers (Wu et al., 2019a;b; Lattner et al., 2020; Liu et al., 2019), fast convergence is desirable for many users of these pipelines to have

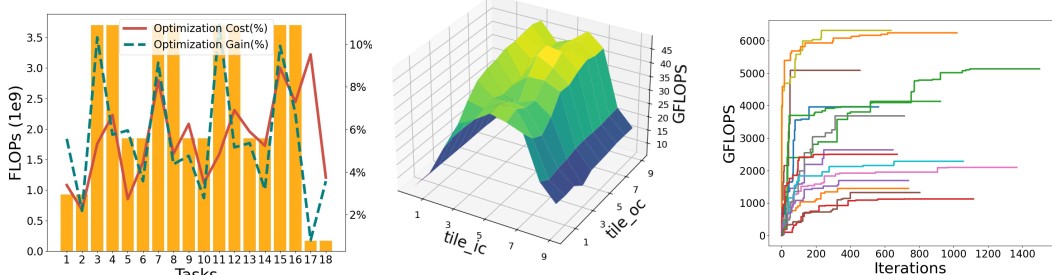

Figure 2: Inproportional optimization gain/cost.

Figure 3: Code transformation space.

Figure 4: DL optimization curves.

a better control of the optimization cost and good performance. For example, deployment engineers may want to obtain an optimized model sooner or quickly get a latency upper-bound estimate of a model in development.

**Challenge 2. Static scheduling has only a limited view of the tensor program and has difficulty taking advantage of the actual optimization behavior.** We note that from an execution point of view, the optimization of tensor operators is independent of each other, so that we may optimize them in any order and even non-consecutively. As a result, dynamic optimization has a big advantage for iterative DL compilation: We can intelligently order the optimization sequence of operators (i.e., scheduling) to accelerate the convergence of the optimization significantly. For example, it would be better to switch to optimizing another operator if we convincingly identify that the other operator has a higher potential. That being said, is it realistic to assume that all the information concerning optimizing the operators is available before the optimization even starts so that we can decide the schedule from the very beginning? Our preliminary analysis indicates that the amount of computation of an operator (known a priori) has a very disproportionate impact on the optimization time and latency reduction. Fig. 2 shows that although operator 17 of VGG (Simonyan & Zisserman, 2015) takes the longest time to optimize, it yields the least amount of latency reduction[1]. Our further investigation shows that the underlying code transformation space is non-linear, as shown in Fig. 3 [2]. As a result, the optimization behavior tends to change over time, which is hard to recognize and predict with static knowledge only.

**Challenge 3. Even with dynamic information, it is not clear how to best extrapolate estimated performance.** Given the optimization results, there is an incentive to adopt a "*predict-then-optimize*" paradigm that builds a model to learn the correlation between the optimization cost and the observed optimization performance the model can be used to make predictions for potential performance gains. To identify the characteristics of the optimization behavior, we plot 16 optimization curves of best-found GFLOPS (Giga Floating Point Operations Per Second) in Fig. 4 to find patterns that can be used for designing a prediction model. We find that most curves (1) roughly follow an increasing curve with a diminishing return, (2) saturate towards an unknown final value, and (3) occasionally exhibit sudden jumps. The curve saturates to an unknown value because the performance cannot exceed the hardware peak GFLOPS, which is 9.7-TFLOPS in our case. The curve has sudden jumps because the code transformation space has change points, as shown in Fig. 3. By taking into account the curve information, we believe it has more opportunity to dynamically optimize operators that likely lead to greater performance improvements.

---

[1]The orange bar shows the amount of computation of each operator measured as the floating-point operations (FLOPs), which can be calculated statically before the optimization starts, as described in Molchanov et al. (2017). The "optimization gain" is calculated as the reduction of wall-clock time from each operator after optimization, and the "optimization cost" is calculated as the wall-clock time spent to obtain the optimized latency, both of which are normalized by the total latency reduction and optimization time.

[2]The figure shows the code transformation space of a Conv2D operator in ResNet-18. In this case, the performance of this operator varies based on the tiling size along the input channel and output channel while having other knobs fixed. The knobs control various aspects of the optimization and its performance. A summary of the knobs can be found in Ahn et al. (2020).

## 4 METHOD

In this section, we present our design for DynaTune. We illustrate the difference between the existing DL optimization (Fig. 5) and the high-level design of DynaTune (Fig. 6).

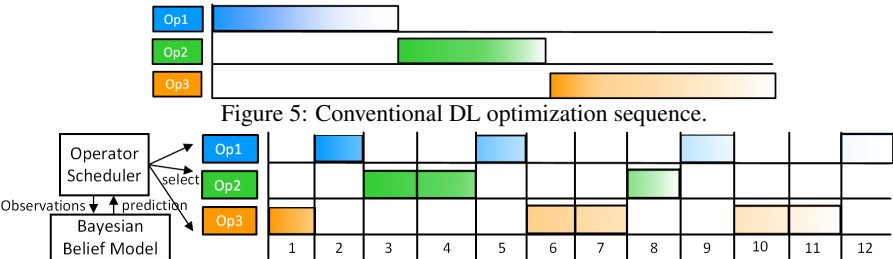

Figure 5: Conventional DL optimization sequence.

Figure 6: DynaTune execution with dynamic optimization and Bayesian belief model.

### 4.1 DYNAMIC MULTI-TENSOR-OPERATOR OPTIMIZATION PROBLEM

In this paper, we take a view of accelerating the convergence speed of *multi-tensor-operator optimization* by first considering a Multi-Armed Bandits model for the problem. In particular, we partition time into some fixed-length time slots $\{1, 2, ..., T\}$. Similar to MAB, we define an *operator scheduler* that operates in discrete time steps. At the beginning of any time slot $t \in \{1, 2, ..., T\}$, the scheduler needs to choose an operator $k_t \in [K]$ for tuning in that slot. The scheduler then obtains a list of observations (i.e., best-found performance measured in GFLOPS) $Perf_k(t \cdot L_k : (t+1) \cdot L_k)$ of $k$, where $L_k$ is the number of iterations an operator has been optimized. The remaining unchosen operators stay the same unless it is selected (i.e., rested arms).

The latency reduction for $k$ at time slot $t$ is $r_t(k_t) = \frac{op(k)}{Perf_k[t \cdot L_k]} - \frac{op(k)}{Perf_k[(t+1) \cdot L_k]}$, where $op(k)$ represents the number of floating point operations of $k$. We further define an optimal schedule,i.e., $\pi^* = \{k_1^*, ..., k_t^*, ...k_T^*\}$, where $k_t^*$ is the operator at step $t$ one could have taken in hindsight (after seeing all performance realizations) that would yield the maximum latency reduction. The cumulative regret is then defined as: R(T) = $\sum_{t=1}^{T}(r_t(k_t^*) - r_t(k_t))$, where $k_t^*$ is the operator at step $t$ one could have taken in hindsight (after seeing all performance realizations) that would yield the maximum latency reduction. The goal is therefore to design the scheduler, such that the cumulative regret is minimized. We call this a *dynamic multi-tensor-operator optimization problem*.

### 4.2 TIME-SLOT-BASED SCHEDULING

To select which operator to optimize in a time slot, we consider deterministic selection policies that belong to the class of index-based MAB policies (Gittins et al., 2011). Among many options, we choose *upper confidence bound* (UCB) as our action function to represent the exploration and exploitation tradeoff (Auer et al., 2002). In particular, assume $c_t[k]$ be the number of times that an operator $k \in [K]$ has been selected for optimization up to slot $t$. If $c_{t-1}[k] > 0$, we denote $y_k(t)$ to be the estimated performance of operator $k$ at the end of $t$. The UCB for operator $k$ at $t$ is defined as $u_k(t) = r_k(t) + \sqrt{C \times \frac{\log t}{c_{t-1}[k]}} = (\frac{op(k)}{Perf_k[(t-1) \cdot L_k]} - \frac{op(k)}{y_k(t)}) + \sqrt{C \times \frac{\log t}{c_{t-1}[k]}}$ for $c_{t-1}[k] > 0$ and $u_k(t) = \gamma$ (constant) for $c_{t-1}[k] = 0$. The scheduler then computes UCB for each operator and selects the next one that maximizes UCB.

Our definition of UCB measures the potential latency reduction from an operator $k$ compared to other operators' expected performances. The first term is a point estimate of the future latency reduction [3], converted from GFLOPS improvement. The second term is related to the size (according to Chernoff-Hoeffding bounds (Hoeffding, 1994)) of the one-sided confidence interval, which allows the true expected performance falls within with overwhelming probability. In the evaluation section, we evaluate several other action functions, including $\epsilon$-greedy and softmax sampling. We

---

[3]In practice, an operator may appear multiple times in a network. Depending on the implementation, the compiler may reuse the same transformation plan for operators that have the same shape. In that situation, we multiply the point estimate of the future latency reduction in the reward function with a weight factor that represents the times of the corresponding operator that appears in the network.

observe that these methods, in general, offer very similar performance. We choose UCB since some recent theoretical studies show that the growth rate of the regret from UCB is sublinear and long-run average optimal in the non-stationary bandit setting (Cortes et al., 2017).

### 4.3 BAYESIAN BELIEF MODELS FOR EXPECTED PERFORMANCE IMPROVEMENTS

To obtain an optimal solution, $k_t^*$ requires perfect information, hence infeasible to achieve in practice. Since we identify some patterns (in Sec. 3) that indicate that there is a functional relationship between the expected performance and the already observed performance, we propose a Bayesian belief model $f_k(t)$, enabled by Markov Chain Monte Carlo (MCMC), to capture our current knowledge of the optimization results to predict future performance and get updated as new observations become available. In particular, we choose parametric curve models whose shape coincides with our knowledge about the form of optimization curves: increasing, saturating functions such as those from the power-law (*pow2*, *pow4*, *log power*) or the logarithmic family (*loglinear*, *logloglinear*). Among these functions, we choose *log power* function as it works well in our case.

To handle breakpoints as mentioned in Sec. 3, we employ a piece-wise parameterization to improve the approximation to the shape of the underlying relationship: since the abrupt jumps often happen just a few times, we cut observation $x$ into segments if a jump leads to more than a relative $\Delta\%$ (i.e., 20% higher GFLOPS) improvement and model segments with different curve parameters.

---

**Algorithm 1**                     **DynaTune: Dynamic Multi-Tensor-Operator Optimization**

---
1: **Input:** A model $\Phi$ with a list of [K] = $\{1,...,K\}$ operators
2: **Output:** An optimized model $\Phi^*$
3: **Init:** $c = (0, 0,...,0)$
4: **for** t = 1,...,T **do**
5:      **for** k = 1,...,K **do**
6:          Observe history performance of operator $k$ and predict a future performance using the belief model in Section 4.3
7:          Update the UCB value $u_k(t)$ using the equation in Section 4.2
8:      $k_t \leftarrow \arg\max u_k(t)$
9:      Allocate time slot $L$ to $k_t$ for actual optimization
10:     $c[k] \leftarrow c[k] + 1$
11:     **if** k.finished **then**
12:         Exclude $k$ from [K]
13:     Collect new data of observed GFLOPS from optimizing $k$ and update the belief model

---

**Data normalization.** Before feeding the observations to the model, since the observed GFLOPS of different operators has different ranges, we normalize the GFLOPS of all operators to a common scale. Given that the maximum GFLOPS of any operator is bounded by the theoretical peak performance on the target deployment hardware (e.g., 9.3-TeraFLOPS on a Nvidia P100 GPU), we apply the following formula to normalize observation $x$ as follows: $Normalized\ x = x/Peak(target\_hardware)$ which transfers the observation's values to a new range 0 to 1. We convert the normalized $x$ back to GFLOPS when calculating UCB.

**Modeling uncertainty.** Since our goal aims at allocating time to optimize tensor operators that are highly likely to bring performance improvement, we need to model uncertainty as truthfully as possible. To model uncertainty, we perform MCMC sampling from the posterior $p(\theta_k|Perf_k[1 : t \cdot L_k]) \propto p(Perf_k[1 : t \cdot L_k]|\theta_k)p(\theta_k)$ of model parameters $\theta_k$ of the curve function $f_k(t)$ given the observed performance $Perf_k[1 : t \cdot L_k]$. A sample approximation for $Perf_k[t']$ with $t' > t$ can then be formed as

$$E[Perf_k[t']|Perf_k[1 : t \cdot L_k]] \approx f_k(t'|\theta_k), \tag{1}$$

Among many options to do MCMC sampling, we choose Goodman & Weare's Affine Invariant MCMC Ensemble sampler (Goodman & Weare, 2010), which significantly outperforms standard M-H methods and produces independent samples (which takes O(N) likelihood evaluations) with a much shorter autocorrelation time. We initialize the ensemble samplers in a tight N-dimensional Gaussian ball in parameter space around the maximum likelihood result, which is obtained through non-linear least-squares fit. We set a uniform prior for $\theta$ and make sure the prediction is non-

decreasing, i.e., the predicted performance is not worse than the more recently observed best-found performance, by also explicitly encoding this knowledge into the prior. We obtain the predictive mean $\mu$ and standard deviation $\sigma$ of the posterior parameter distribution using 200 MCMC samples. We then compute the *expected positive improvement* EI (Brochu et al., 2010) over the best known measured performance, while taking into account the possible uncertainty in that prediction.

# 5  EVALUATION

In this section, we evaluate DynaTune experimentally, seeking answers to how DynaTune helps accelerate the convergence of optimizing all operators in a model. We integrate DynaTune with AutoTVM (Chen et al., 2018b) and use as AutoTVM our baseline for comparison. We implement DynaTune in Python, and we leverage emcee (Foreman-Mackey et al., 2013) to implement the MCMC sampling. We do a warmup to find the first non-zero GFLOPS as a starting point for all operators. We use the default hyperparameters provided by AutoTVM for the underlying code optimization. To obtain the parameter posterior, we run the ensemble MCMC with 10 walkers and 500 sampling steps. A convergence diagnostics of MCMC is presented in Appendix B. All the free parameters in the curve model are taken care of by MCMC sampling. For UCB, we choose a default value of $C = 2$ suggested by the theory in Auer et al. (2002), which we find to be robust to different range of latencies. When the initial latency is <1ms, we empirically find that C=0.2 leads to increased performance, which we report. We perform 5 independent runs of each configuration with different random seeds and report the median together with a 95% confidence interval.

## 5.1  COMPARISON OF AUTOTVM AND DYNATUNE FOR OPTIMIZING MODELS WITH MULTIPLE TENSOR OPERATORS

In the previous approach (Chen et al., 2018b), authors optimize one operator at a time until all operators in a model have been optimized. We compare the performance of AutoTVM and DynaTune on how much optimization speedup we obtain as a function of the wall-clock time. Due to space limitations, we include four tasks, covering both CPU and GPU hardware: ResNet-18 (He et al., 2016) and SqueezeNet (Iandola et al., 2016) on CPU (Intel Xeon CPU E5-2690 v3 @ 2.60GHz 2600 MHz), VGG (Simonyan & Zisserman, 2015) Transformer Encoder (Iandola et al., 2016) on GPUs (Nvidia Tesla P100), which have K = 12, 18, 18, and 6 tunable operators, respectively.

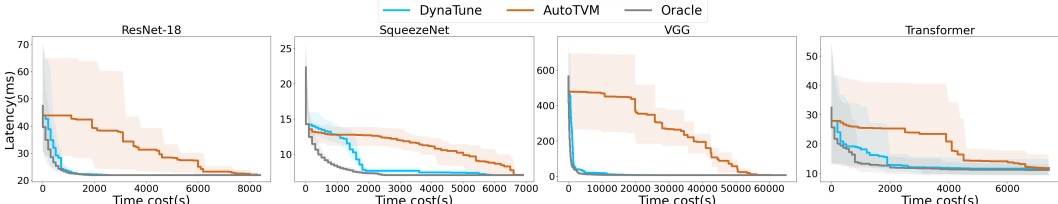

Figure 7: Comparison of the convergence speed and quality among DynaTune, AutoTVM, and Oracle on both CPU and GPUs.

Fig. 7 visualizes the results. The x-axis denotes the wall-clock time of optimizing the model. The y-axis denotes the lowest model latency obtained as time moves on. Overall, the observation is that DynaTune's dynamic optimization converges significantly faster than the baseline most of the time and is 1.2–2.4 times faster than the baseline to achieve the lowest latency. The baseline has a much slower convergence, because it tries to find optimal transformation plan for one operator before starting optimizing another one. Since there can be an order of magnitude difference between the optimized and unoptimized code, the model latency remains high until the last operator has been optimized. In contrast, DynaTune is able to expedite the optimization significantly by reordering the optimization sequence of operators and dynamically pick promising operators to optimize. As a result, DynaTune obtains the same optimization quality as the baseline but in a much faster speed. Furthermore, we also plot the optimization by assuming having access to the oracle information. For ResNet-18 and VGG, DynaTune gets optimization results close to the oracle. For SqueezeNet and Transformer, DynaTune converges slower than the oracle in the beginning but quickly catches up at around one third of the optimization time, presumably because it is more difficult for the Bayesian model to predict performance in the beginning then later, indicating room for improvement. These

results confirm that a dynamic approach like DynaTune is capable of performing model-level code optimization in a much more efficient way than the existing approach.

## 5.2 COMPARISON OF DYNATUNE WITH STATIC SCHEDULES

In this section, we compare the effectiveness of DynaTune with static allocation schemes. In particular, we compare with three mechanisms: (1) Random, which randomly assign time slots to operators, (2) Round-robin, which assigns time slots in circular order, and (3) Linear, which allocate time linearly with respect to the number of floating-point operations each operator has. Fig. 8 shows that DynaTune consistently outperforms other static schemes and achieves 1.1–2.4 times speedup to obtain the lowest latency. As mentioned in Sec. 3, static knowledge alone is insufficient for making well-suited schedule decisions. In contrast, the improvement in DynaTune comes from constantly making decisions and replaning the allocation strategy based on new observations.

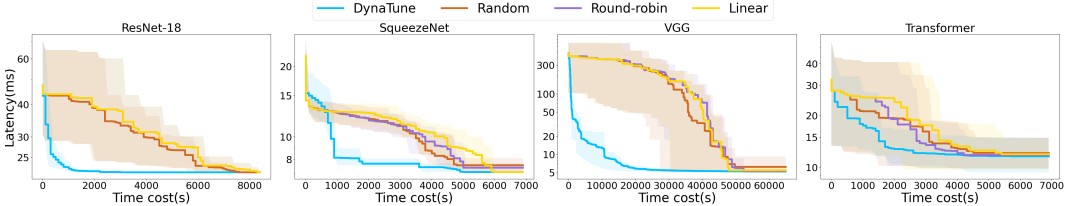

Figure 8: Comparison of convergence speed and quality between DynaTune and static schemes.

## 5.3 COMPARISON OF DYNATUNE WITH ALTERNATIVE DYNAMIC SCHEMES

We also compare the effectiveness of our approach by comparing the following dynamic schemes: (1) Dynamic allocation (DA) + Random selection (Rand): Randomly assigns time slots to operators and reallocates a time slot if the chosen operator has been early terminated. (2) Dynamic allocation (DA) + Round-robin selection (RR): Assigns time slots in circular order, and keep other settings the same as the "DA + Rand" configuration. For all schemes, we use the same early termination condition, e.g., an operator is considered as finished if it has not seen any GFLOPS improvement for a given number of iterations (e.g., 200). Fig. 9 shows that, when not equipped with the Bayesian model (e.g., Rand and RR), the optimization converges relatively slower than with the model to obtain a lower latency. In contrast, DynaTune achieves 1–1.4 times speedup to obtain the lowest latency, presumably because the prediction helps to identify operators that potentially bring high latency reduction. We also evaluated DynaTune with two alternative action functions: $\epsilon$-greedy and softmax sampling. As Fig. 14 shown, while UCB offers marginal gains in some cases, the performance of all three action functions are very similar (more results can be found in Appendix A), indicating that potentially all three action functions can be used for operator selection.

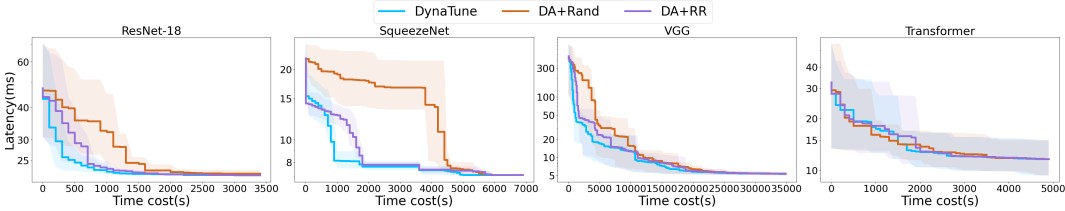

Figure 9: Comparison of convergence speed and quality between DynaTune and alternative dynamic schemes.

## 5.4 COMPARISON WITH CHAMELEON

Chameleon (Ahn et al., 2020) is a recent work that uses reinforcement learning and adaptive sampling to efficiently explore the code transformation space of a tensor operator. In this part, we compare DynaTune with the open-sourced version of Chameleon[4]. We follow the provided instructions and evaluate Chameleon on the same set of models: ResNet-18 and SqueezeNet on CPU, VGG and Transformer on GPU. Figure 10c and Figure 10d show that although Chameleon is faster than

---

[4]https://github.com/anony-sub/chameleon

the baseline AutoTVM on VGG and Transformer on GPU, it is slower than DynaTune to converge to the lowest latency. This is because although the optimization speed of each operator has been improved by Chameleon, it still sequentially optimizes one operator at a time. Therefore, the overall convergence speed is still bounded by the least optimized tensor operators. In contrast, DynaTune focuses on improving the convergence speed of multi-tensor-operators and dynamically allocates time to improve the overall convergence speed.

We also evaluate Chameleon on CPU, given that model inference is not uncommon on CPU (Wu et al., 2019a; Zhang et al., 2019; 2018). When running on CPU, a bit surprisingly, we observe that Chameleon is slower than the baseline AutoTVM on ResNet-18 (Figure 10a) and SqueezeNet (Figure 10b). We analyze the performance and find that the RL optimizer in Chameleon adds a non-trivial amount of overhead than the default optimizer (SA + XGBoost) used in AutoTVM on CPU. As a result, although Chameleon reduces the hardware measurement cost and the number of iterations to find the best configuration, its optimization time on CPU is longer than the baseline AutoTVM because of this extra overhead and is therefore also slower than DynaTune on CPU. Overall, DynaTune is 1.4–4.7 times faster than Chameleon to reach the same latency. Although DynaTune is faster than Chameleon, we want to point out that DynaTune can be combined with Chameleon to achieve better performance, at least on GPU.

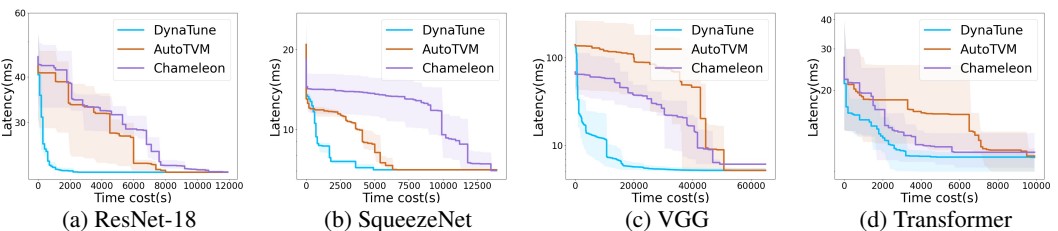

(a) ResNet-18      (b) SqueezeNet      (c) VGG      (d) Transformer

Figure 10: Comparison with AutoTVM, Chameleon, and DynaTune.

## 5.5 MORE ANALYSIS RESULTS

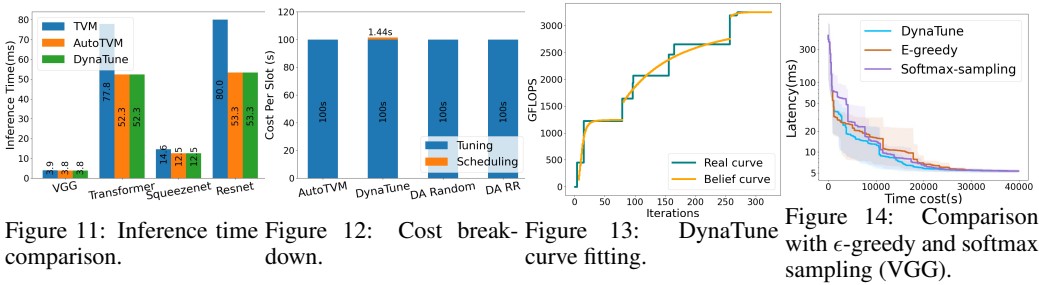

Figure 11: Inference time comparison.    Figure 12: Cost break-down.    Figure 13: DynaTune curve fitting.    Figure 14: Comparison with $\epsilon$-greedy and softmax sampling (VGG).

**Inference time comparison** Fig. 11 compares the final inference time optimized by TVM, AutoTVM, and DynaTune respectively. Overall, DynaTune achieves up to 26.7% faster inference speed over TVM and similar code performance as AutoTVM. AutoTVM and DynaTune achieve much higher speedup on ResNet-18 and Transformer, presumably because the heuristic-based optimizations in TVM are sub-optimal. While achieving comparable optimization quality as AutoTVM, DynaTune significantly reduces the lengthy optimization time.

**Segment curve fitting.** Fig. 13 gives an example on the curve fitting. It is easy to see that the data between 1 and 80 could be approximated well by one segment, and the data between 80 and 260 could be approximated by another segment.

**Cost breakdown.** Fig. 12 shows the breakdown in the average time required scheduling a time slot (averaged across four models), which involves the optimization time (e.g., 100s) and the scheduling cost. Overall, baseline (sequential), random, and round-robin incur almost no overhead in scheduling. Our Bayesian belief model takes time to do fitting (MLE and MCMC) and inference. However,

it only adds 1.44s, which adds 1.4% overhead and can be easily compensated by the time saved from picking a promising operator to optimize.

## 6 Related Work

DynaTune uniquely offers a solution that exclusively enables (i) Multi-Armed Bandits based dynamic multi-tensor-operator optimization and (ii) extrapolation of optimization performance by Bayesian inference in the context of (iii) optimizing DL compilers. As such, we discuss the related work from each of the three independent research directions.

**Optimizing compilers.** TVM (Chen et al., 2018a) and TensorComprehensions (Vasilache et al., 2018) use simulated annealing and genetic algorithm to search the best code transformation plan for neural networks. Subsequently, AutoTVM (Chen et al., 2018b) incorporates boosted decision trees as a surrogate model to reduce the number of real hardware measurements. CHAMELEON (Ahn et al., 2020) proposes to use reinforcement learning for efficient search space exploration. AdaTune (Li et al., 2020) cuts the cost of hardware measurement through adaptive evaluator and allows to better adapt optimizations against hardware and model heterogeneity. However, while existing approaches study how to accelerate the convergence of individual operators, DynaTune focuses on improving the convergence speed of the entire model using Multi-Armed Bandit learning for dynamic optimization cross all tensor operators. DynaTune can be combined with other approaches that speed up the convergence of individual operators to maximize gains. Program autotuning has also been studied in the context of generic programming, where black box optimization (Ansel et al., 2014) and runtime information (Novillo, 2014) are utilized to generate optimized code. Rather than advancing the generic problem of program auto-tuning, our technique is designed and validated specifically for the DL model compilation problem.

**Multi-Armed Bandits.** The Multi-Armed Bandits (MAB), first introduced by Robbins (Robbins, 1952), with various modifications have been used to model a plethora of dynamic optimization problems under uncertainty (Zelen, 1969; Bergemann & Välimäki, 1996; Kleinberg & Leighton, 2003; Awerbuch & Kleinberg, 2004; Bergemann & Hege, 2005; Caro & Gallien, 2007; Pandey et al., 2007). To the best of our knowledge, our work exclusively explores a different problem, which is optimizing DL compilers using MAB. Furthermore, our problem can be viewed as an instance of a *rested, non-stationary* MAB, which is complicated to solve since the evolution of the stochastic process depends on the choices made by the algorithm. We are able to design new UCB-style algorithms that improve the DL compilation in this specific non-stationary rested bandit scenario.

**Bayesian Inference.** Bayesian inference is a broad field and has been widely used in statistical analysis (Box & Tiao, 2011). For example, Bayesian inference has been used in hyperparameter tuning (Domhan et al., 2015; Lu et al., 2019). DynaTune shares similarities with these methods in building a Bayesian model for prediction, but it differs in its context and has its unique challenges. The Bayesian belief model we design is to predict operators that may lead to greater latency reduction whilst performing an optimization to accelerate the process.

## 7 Conclusion

Although highly optimized code can be achieved through existing DL compilers, an obvious drawback is that they optimize operators one at a time, leading to slow convergence of optimization speed when the model has multiple operators. In this paper we have introduced a method, called DynaTune, which treats the optimization of multiple operators in a model as a whole and dynamically optimizes all operators to expedite convergence. Combined with a Bayesian belief model, the dynamic optimization prioritizes operators that have larger latency reduction. As a result, DynaTune achieves much faster convergence speed in getting optimized models, outperforming the state-of-the-art approaches.

### Acknowledgement

The authors appreciate the anonymous ICLR reviewers for providing constructive feedback for improving the quality of this paper. All authors are not funded by any other agency.

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

## A    COMPARISON WITH $\epsilon$-GREEDY AND SOFTMAX SAMPLING

We compare DynaTune with two alternative operator selection methods: (1) $\epsilon$-greedy: Like our approach, but UCB is replaced with $\epsilon$-greedy, which select the operator with the highest expected latency reduction based on EI with probability $1 - \epsilon$ (e.g., $\epsilon$=0.05) and randomly select an operator with $\epsilon$ to ensure exploration. (2) Softmax: Like our approach but UCB is replaced with softmax sampling, which stochastically samples operators to optimize based on expected latency reduction at each step. Fig. 15–Fig. 18 show the comparison results. With the Bayesian model enabled in cases, we observe that softmax sampling, $\epsilon$-greedy, and UCB work very similarly across multiple models. We choose UCB in our design because it has been theoretically studied that UCB has a sublinear regret growth rate.

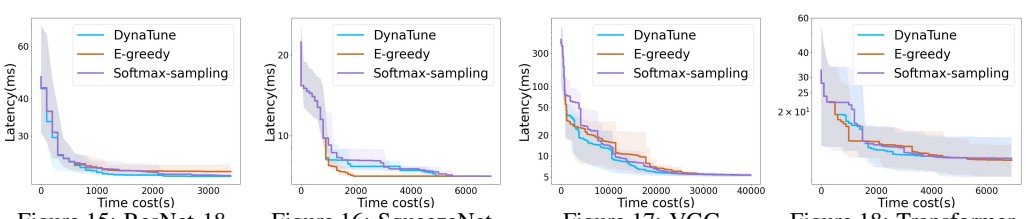

Figure 15: ResNet-18.      Figure 16: SqueezeNet.      Figure 17: VGG.      Figure 18: Transformer.

## B    MCMC ANALYSIS

Fig. 19 plots the time series (i.e., trace) of the parameters in the Markov chains to assess the behavior of our MCMC sampling. It shows the positions of each walker (10) as a function of the number of steps in the chain. The walkers start in small distributions around the maximum likelihood values and then quickly wander and start exploring the full posterior distribution. In fact, after about 100 steps, the MCMC appear to have converged to stationarity (i.e., self-replicating). We also examine

the mean acceptance fraction of the ensemble, which is 0.54 by average, and it indicates that the MCMC sampler has generated a sufficient number of parameter samples that have also gained good information about the underlying distribution.

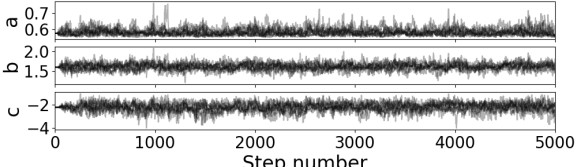

Figure 19: Ensemble MCMC trace for $\theta$ (a,b,c).

## C  COMPARISON WITH ANSOR SCHEDULER

In this section, we compare our approach with a concurrent work called Ansor (Zheng et al., 2020). We use the default hyperparameters $\alpha = 0.2, \beta = 2$, backward_window_size=3 in the open-sourced implementation [5] for the evaluation. We use the same evaluation methodology as ours by performing five independent runs of each configuration with different random seeds and reporting the median together with a 95% confidence interval. Figure 20a–20d show that both DynaTune and the Ansor scheduler outperform the baseline AutoTVM by a large margin. This is expected, because both approaches accelerate the convergence of model optimization through dynamic optimization. Figure 21a–21d show a more detailed comparison between DynaTune and Ansor. Overall, Ansor seems to reduce the latency faster in the beginning, but DynaTune always catches up and is often quicker to reach the lowest latency. For example, DynaTune achieves a faster convergence to reach the lowest latency than Ansor on ResNet-18, VGG, and Transformer.

Although Zheng et al. (2020) and our approach share a similar high-level idea of dynamically allocating time resources to different tasks, the exact mechanisms on how to allocate resources are very different. The tasks scheduler from Zheng et al. (2020) decides which task to optimize based on a heuristic score, which is a weighted sum between the latency reduction rate of a task in a recent small time window and an expected latency reduction in the future based on task similarity information. The estimation of the score not only requires defining and heuristically adjusting similarity groups but also requires two hyperparameters $\alpha$ and $\beta$ to control the weight to decide which estimations to trust more. However, it is not immediately clear how such weights should be set and how to adapt them to different models. For example, the default hyperparameter settings may cause the scheduling to be overly greedy (i.e., getting stuck at a local optimum), which may explain why Ansor converges faster in the beginning but is slower to reach the best latency towards the end. In contrast, we use a Bayesian belief model to predict how much expected latency reduction from each task in the next time slot, and all the free parameters in our belief model are taken care of by MCMC sampling. Furthermore, our approach takes uncertainty quantification into account, which presumably helps the search escape from local optimum.

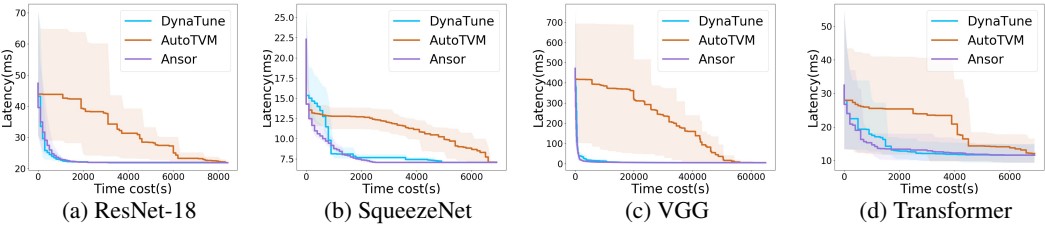

|  (a) ResNet-18 | (b) SqueezeNet | (c) VGG | (d) Transformer |

Figure 20: Comparison with AutoTVM and Ansor scheduler.

[5]https://github.com/apache/incubator-tvm/blob/main/python/tvm/auto_scheduler/task_scheduler.py

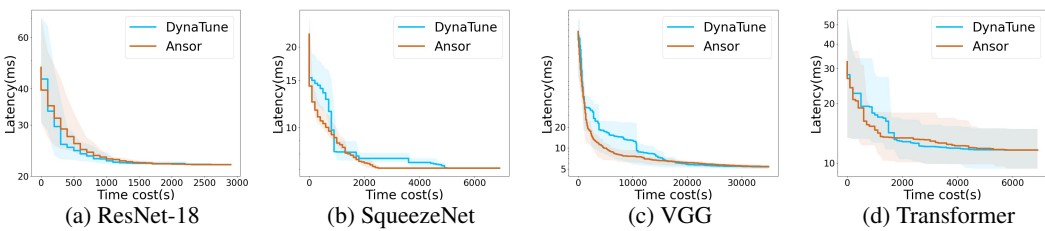

(a) ResNet-18      (b) SqueezeNet      (c) VGG      (d) Transformer

Figure 21: Comparison with AutoTVM and Ansor scheduler.

