# OpenReview forum: "DynaTune: Dynamic Tensor Program Optimization in Deep Neural Network Compilation"
_ICLR.cc/2021/Conference — ICLR 2021 Poster_

### Official Review · AnonReviewer2 · 2020-10-27
**Official Blind Review #2**

**Rating:** 7
**Confidence:** 4

**Review:**

The paper tackles the optimization speed of the deep neural networks. The paper models the operator optimization scheduling as a Multi-Armed Bandit problem. DynaTune first selects the operation to optimize in a discrete slot of time using a Bayesian Belief Model, then optimizes the model given the slot of time.

This interesting paper may help expedite the compilation time of the deep neural networks. As such, this paper tackles on a very important problem. I view this paper as a lineage of papers on "Expedited Compilation" and I believe this fresh view of the problem will open up more possibilities for exploration.

Modeling uncertainty to understand the gains of optimizing a particular operator is very interesting, and the way it leverages this to schedule the optimization process itself is novel.

It seems that the approach can be generically combined with different optimization frameworks such as genetic algorithms. It would be nice to have such discussion and would provide interesting insights.

As a side note, however, there was a paper the tackled the optimization of the inference speed of the end-to-end network [1] which is also applied to TVM. It would be interesting to see how the overall latency achieved by DynaTune compares to this approach.

Questions:
1. How does the end-to-end latency compare to [1]?
2. How are the results affected when put together with GA and Random Search for the actual optimization (line 6, 9 of Algorithm 1)

[1] "Optimizing CNN Model Inference on CPUs", ATC 2019

---

> ### Author Response · Authors · 2020-11-19
> **Response to Reviewer #2**
>
> Thank you for your positive feedback! We address the comments and feedback you provided below.
>
> **Q1: How does the end-to-end latency compare to [1]?**
>
> A1: NeoCPU [1] studies how to get the best performance of convolution operators on CPU. DynaTune is complementary to [1], as it proposes a general technique to accelerate the optimization speed of multiple tensor operators by better allocating time resources. It also looks beyond CNN workloads and CPU and targets more diverse network structures (e.g., Transformer networks) and heterogeneous hardware (e.g., CPU and GPU). Moreover, both NeoCPU and DynaTune are built on top of the open source TVM project. The optimizations in NeoCPU, such as data layout transformation and vectorization, had been merged into the target-dependent passes of TVM, whereas the optimization of DynaTune is at the Learning-to-Compile stage (please kindly refer to the background section and Figure 1 to see their place in the DL compilation pipeline). The end-to-end latency of TVM is already a result of applying the optimizations from [1]. Figure 10 in the manuscript reports the end-to-end inference time comparison between TVM and DynaTune for SqueezeNet and ResNet on CPU. DynaTune is **16\% and 50\% faster** than TVM on SqueezeNet and ResNet, respectively.
>
> **Q2: How are the results affected when put together with GA and Random Search for the actual optimization (line 6, 9 of Algorithm 1)**
>
> A2: We thank the reviewer for the comment about the synergy between DynaTune and alternative optimization methods. We use simulated annealing (SA) and XGBoost based statistical cost model to optimize individual operators. We choose SA + XGBoost in our experiments because [2] shows that SA + XGBoost is more effective than random search and genetic algorithms for tensor optimization (shown in Figure 4 in [2]). For this reason, SA + XGBoost is chosen as the default optimization scheme in AutoTVM. However, the reviewer is correct that DynaTune can be combined with alternative mechanisms beyond SA + XGBoost for the actual optimization to achieve even better performance. This would make an interesting future study.
>
> [1] Liu et al. "Optimizing CNN Model Inference on CPUs", https://arxiv.org/abs/1809.02697
> [2] Chen et al., "Learning to Optimize Tensor Programs", https://arxiv.org/pdf/1805.08166.pdf
>
> We hope our response has mostly addressed your questions. We are glad to continue the discussion to address any remaining questions you may have.

---

> > ### Comment · AnonReviewer2 · 2020-11-19
> > **Comments**
> >
> > Regarding Q2, from experiences in running numerous experiments, there were situations where running many steps of GA (beyond 800-1000 steps, something like 3000-4000 steps) would yield better results than running SA with XGBoost. This seemed (my speculation) to be due to some overfitting issue of the XGBoost cost models. As this was the case, I am hoping that using DynaTune to enable more steps of optimization may present some interesting results. I believe such results would be interesting addition to the paper.
> >
> > Overall, I am satisfied with the authors' response. I really hope to see this paper in the conference as I believe this paper presents an interesting application of ML (Bandits) in the context of Systems that may have practical impact in ML compilers.

---

### Official Review · AnonReviewer3 · 2020-10-28
**The paper approaches DNN compilation by formulating it as a Multi-armed bandit problem. Interesting challenges to DNN compilation are identified, a sound methodology is proposed and experimentally validated.**

**Rating:** 7
**Confidence:** 1

**Review:**

The paper investigates the use of online optimization to improve the performance in Deep Neural Network (DNN) compilation.
The three drawbacks of the existing approach/challenges in DNN compilation highlighted in the paper are (1) optimizing individual tensor operator performance instead of that of the entire model, (2) Static scheduling is oblivious to the optimization behavior, (3) extrapolating estimated performance while tuning.

The authors model the model as scheduling in a time-slotted system, where they model the average performance (mean reward) of each operator using some performance curve obtained empirically. These curves capture the variability of operators across training duration, which helps with (1) and (2). For tackling (3),  the variability of the performance (noise) of each operator is estimated using some MCMC sampling techniques. Finally, using the mean reward and noise they cast the problem as finding the expected optimal schedule - a sequence of operators to be chosen during the training, as a Multi-armed bandit problem. The authors then design the DynaTune Algorithm base on a well-known UCB algorithm to solve this problem.

The experiments show DynaTune outperforms AutoTVM (Chen et al., 2018b) the apparent state-of-the-art (see, disclaimer below) in many important DL architectures.


Pros:
The problem of optimizing DL architectures is an important problem, as they are ubiquitous. The three challenges mentioned here sound credible (to an outsider), and in gains of dynamic scheduling is well known. Overall, the problem is well-motivated, and the solution proposed is justified. The experiments seem to give DynaTune an edge over an existing method.



Cons:
* How Fig 2,3,4 are obtained is not clear to me. Some pointers to empirical studies or the methodology used are necessary.

* The details behind the variability estimation using the MCMC method is also not clearly written.

* The baseline AutoTVM (Chen et al., 2018b) is somewhat old. However, I am not sure if a better baseline is available.







Disclaimer: Although I am familiar with works on Bandits, I am a complete outsider to the topic of DNN compilation.

---

> ### Author Response · Authors · 2020-11-19
> **Response to Reviewer #3 -- Part 1**
>
> We thank Reviewer#3 very much for the positive feedback and for highlighting the significance of our work.
>
> **Q1: How Fig 2,3,4 are obtained is not clear to me. Some pointers to empirical studies or the methodology used are necessary.**
>
> A1: Here, we would like to address the reviewer's concern by providing the details on how Figure 2,3,4 are obtained with some pointers. Figure 2 shows the analysis results of 18 tunable operators in the VGG model on GPU. The amount of computation of each operator is measured as the floating-point operations (FLOPs), which can often be calculated based on the shape and dimensionality of an operator, e.g., a matrix-multiplication [M, K] x [K, N] has 2 x M x K x N floating-point operations. For FLOPs calculation of Conv2D, the reviewer can refer to A.1 in https://arxiv.org/pdf/1611.06440.pdf. The compilation framework TVM we use automatically calculates FLOPs for each operator. The "optimization gain" is calculated as the reduction of wall-clock time from each operator, and the "optimization cost" is calculated as the wall-clock time spent to obtain the optimized latency. They are normalized by the total latency reduction or optimization time in Figure 2.
>
> Figure 3 shows the code transformation space of a Conv2D operator in ResNet-18 on CPU. In this case, the performance of this operator varies based on the tile size (a code optimization technique) along the input channel and output channel while having other knobs fixed. The knobs control various aspects of the optimization and determine whether the code (1) fully utilizes the internal parallelism within processors, (2) uses the shared memory wisely, and (3) maximizes data locality. A summary of the knobs can be found in Table 1 in https://arxiv.org/pdf/2001.08743.pdf.
>
> Figure 4 plots the optimization curves of all operators in ResNet-18 using AutoTVM on Nvidia Tesla P100 GPU. The x-axis represents the number of optimization iterations, and the y-axis represents the best-seen GFLOPS (Giga Floating Point Operations Per Second) at each iteration step of a task.
>
> **Q2: The details behind the variability estimation using the MCMC method is also not clearly written.**
>
> A2: We thank the reviewer's comment. Appendix B has a more detailed analysis of the MCMC method. Figure 18 plots the time series of the parameters in the Markov chains to assess the behavior of our MCMC sampling. We also report the mean acceptance fraction of the ensemble MCMC sampler, which indicates that our MCMC sampler has generated a sufficient number of parameter samples that have gained good information about the underlying distribution. We can provide more analysis results per the reviewer's requests.

---

> > ### Author Response · Authors · 2020-11-19
> > **Response to Reviewer #3 -- Part 2**
> >
> > **Q3: The baseline AutoTVM (Chen et al., 2018b) is somewhat old. However, I am not sure if a better baseline is available.**
> >
> > A3: We understand the reviewer's concern. There have been very few attempts to utilize a scheduler to simultaneously optimize multiple tensor operators in deep neural networks. To relax the reviewer's concern, we added additional experiments on Chameleon from Ahn et al. (https://arxiv.org/pdf/2001.08743.pdf) in Appendix C. Chameleon, which we cited and discussed in the related work section, is a recent work that uses reinforcement learning and adaptive sampling to accelerate the search space exploration of a tensor operator, but it still optimizes one operator at a time. Therefore, although the optimization speed of each operator has been improved, the overall convergence speed is still bounded by the least optimized tensor operators. We did not compare with Chameleon in the original draft because we considered DynaTune as a technique that was compatible and complementary with existing DL compilation techniques that focused on optimizing individual operators.
> >
> > We use the source code of Chameleon from https://github.com/anony-sub/chameleon. We follow the provided instructions and evaluate Chameleon on the same set of models: ResNet-18 and SqueezeNet on CPU, VGG and Transformer on GPU. Figure 19c and Figure 19d in Appendix C of the revised manuscript show that although Chameleon is faster than the baseline AutoTVM on VGG and Transformer on GPU, it is much slower than DynaTune to converge to the lowest latency. When running on CPU, we observe that Chameleon is surprisingly slower than the baseline AutoTVM on ResNet-18 (Figure 19a) and SqueezeNet (Figure 19b). By checking the results in the Chameleon paper, we find that Chameleon seems to have only been evaluated on GPU. We analyze the performance and find that the RL optimizer in Chameleon adds a non-trivial amount of overhead than the default optimizer (simulated annealing + XGBoost cost model) in AutoTVM on CPU. As a result, although Chameleon reduces the hardware measurement cost and the number of iterations to find the best configuration, its optimization time on CPU is longer than the baseline AutoTVM and is much slower than DynaTune on CPU.  Overall, DynaTune is **1.4--4.7 times faster** than Chameleon to reach the same latency. Although Dynatune is faster than Chameleon, we want to point out that DynaTune can be combined with Chameleon to achieve better performance, at least on GPU.
> >
> > We hope we have addressed most of your concerns. Please feel free to let us know if you have any other questions.

---

> > > ### Comment · AnonReviewer3 · 2020-11-20
> > > **I maintain my score as my concerns are addressed**
> > >
> > > The authors have addressed my concerns quite extensively. I will maintain my score. Adding elements from the response to the paper (maybe in the supplementary part) will strengthen the paper, in my opinion.

---

### Official Review · AnonReviewer1 · 2020-10-28
**Nice idea. But lacking a comparision with a highly related paper.**

**Rating:** 6
**Confidence:** 5

**Review:**

##########################################################################

Summary:

The paper proposes an algorithm to optimize the auto-tuning time for compiling neural networks.
 It dynamically allocates time to different operators with a multi-armed bandit algorithm and a Bayesian belief model.
The evaluation results show the proposed method can achieve significant speedup.

##########################################################################

Reasons for score:

I lean to rejection because of some flaws in the paper.
1. It does not mention a highly related paper that follows the same high-level idea (i.e., dynamically allocating time resource). The section 6 task scheduler in this paper (https://arxiv.org/abs/2006.06762) proposes a heuristic-based algorithm for dynamically allocating time resource. It also predicts room for improvement of each task. It supports user-defined objective functions and utilizes similarity between tasks. The authors should at least include a comparison with this method. The code of this method is merged into the TVM project (https://github.com/apache/incubator-tvm/blob/main/python/tvm/auto_scheduler/task_scheduler.py), which is the baseline of DynaTune.
2. The formulation of reward can be improved
In section 4.1, the authors define the reward as the latency reduction of one operator. However, an operator can appear multiple times in a network. So a weight should be multiplied to the reduction. The weight of an operator is the number of the appearance of it in the network.
3. Some assumptions can be improved
In challenge 2 of section 3, the authors claim "We note that the optimization of tensor operators is independent of each other". This is not true because different operators can share a cost model. They can contribute training data to the shared cost model. The shared cost model then influences the search of all operators. By using a shared cost model and transfer learning, the search time can be greatly reduced (as shown in AutoTVM and Ansor), so we cannot say the optimization of different operators is independent. How to model this effect is very challenging.


##########################################################################

Pros:

- Overall, the observations and ideas are correct. The paper is well-written with necessary background information.
- The proposed methods are all reasonable
- The evaluation results are satisfactory and well structured. It correctly evaluates all aspects that I want to know.

##########################################################################

Suggestion for improvement:

The authors can improve their formulation and add a comparison with the related paper I mentioned.

---

> ### Author Response · Authors · 2020-11-19
> **Response to Reviewer #1 -- Part 1 (On lacking a comparison with a highly related paper)**
>
> We thank Reviewer#1 very much for the insightful comments. Our detailed response is listed below.
>
> **Q1: "Lacking a comparison with a highly related paper."**
>
> A1: We appreciate the reviewer's comments. It seems the reviewer's main concern is the lack of comparisons against a method from Zheng et al., recently published in OSDI 2020. OSDI'20 is just held on November 4-6, which is just a few days before today. We thank the reviewer for sharing this latest news, and we will cite it. But we would like you to reconsider using that reason to reject our paper, because the ICLR reviewer instructions (https://iclr.cc/Conferences/2021/ReviewerGuide#faq) explicitly mention that "Authors are encouraged to cite and discuss all relevant papers, but they may be excused for not knowing about **papers not published in peer-reviewed conference proceedings or journals**." Now, we check the code for that paper on Github suggested by the reviewer: \url{https://github.com/apache/incubator-tvm/commits/main/python/tvm/auto_scheduler/task_scheduler.py}. The commit history shows that it was checked in on **October 18th**. That was **after** this manuscript was submitted to ICLR (October 2nd).
>
> That being said, to relax the reviewer's concern, we worked on providing a comparison result with that approach. We uploaded a revised version of the manuscript, in which we added the comparison results to Appendix D. However, we hope to reach the consensus that the evaluation of the merit of this manuscript should not be based on that new experiment. For the evaluation, we use the implementation from https://github.com/apache/incubator-tvm/blob/main/python/tvm/auto_scheduler/task_scheduler.py. The original Ansor paper does not seem to describe how to set the hyperparameters for its scheduler, so we use the default hyperparameters $\alpha=0.2, \beta=2$, backward_window_size=3 in the code. We use the same evaluation methodology as ours by performing five independent runs of each configuration with different random seeds and reporting the median together with a 95\% confidence interval. Figure 20a-20d in Appendix D of the revised manuscript show that both DynaTune and the Ansor scheduler outperform the baseline AutoTVM by a large margin. This is expected, because both approaches accelerate the convergence of model optimization through dynamic optimization. We also added Figure 21a-21d to show a more detailed comparison between DynaTune and Ansor. Overall, Ansor seems to reduce the latency faster in the beginning, but DynaTune can always catch up and is often quicker to reach the lowest latency. For example, DynaTune achieves a faster convergence to reach the lowest latency than Ansor on ResNet-18, VGG, and Transformer.
>
> After reading the Ansor paper in more detail, we find that although the Ansor scheduler from Zheng et al. contains a similar high-level idea of "dynamically allocating time resources to different tasks", the exact mechanism on how to allocate resources is very different from ours. The Ansor task scheduler decides which task to optimize based on a heuristic score, which is a weighted sum between the latency reduction rate of a task in a recent small time window and an expected latency reduction in the future based on task similarity information. The estimation of the score not only requires defining and heuristically adjusting similarity groups but also requires two hyperparameters $\alpha$ and $\beta$ to control the weight to decide which estimations to trust more. However, it is not immediately clear how such weights should be set and how to adapt them to different models. For example, the default hyperparameter settings may cause the scheduling to be overly greedy (i.e., getting stuck at a local optimum), which may explain why Ansor converges faster in the beginning but is slower to reach the best latency towards the end. In contrast, we use a Bayesian belief model to predict how much expected latency reduction from each task in the next time slot, and all the free parameters in our belief model are taken care of by MCMC sampling, which gives our method some plausible advantage in terms of simplicity. Furthermore, our approach takes uncertainty quantification into account, which presumably helps the search escape from local optimum. Apart from these major differences, our manuscript also exclusively provides detailed analysis on the issue of improportional optimization cost/gain (Fig 2), the characteristics of the code transformation space (Fig 3), the shape and pattern of DL code optimization curves (Fig 4), and the connection between tensor compilation and MABs, which we believe would be helpful to provide insight to the community as to why dynamic tensor compilation is needed and what its challenge is.
>
> Given these facts, in our humble opinion, the reviewer may want to reconsider the position on the paper and not discourage the publication of original ideas that have been developed independently and concurrently.

---

> > ### Author Response · Authors · 2020-11-19
> > **Response to Reviewer #1 -- Part 2**
> >
> > **Q2: The formulation of reward can be improved in section 4.1**
> >
> > A2: We appreciate the reviewer's comment. Reusing the same optimization plan for operators that have the same shape is an implementation-level detail, and we believe our main results and conclusion still hold against the multi-tensor-operator optimization problem we formulate in Section 4.1. That being said, this is a valid design point in practice, and we may obtain even better end-to-end latency by incorporating this optimization. Moreover, this optimization should also be easy to incorporate in our design, e.g., by multiplying the point estimate of the future latency reduction in the reward function with a weight that represents the times of the corresponding operator appears in the network. We will consider incorporating it.
> >
> > **Q3: "Some assumptions can be improved in challenge 2 of section 3".**
> >
> > A3: Regarding the "dependency among tensor operators", we revised the text in Section 3 to be more accurate. From an execution point of view, the tuning of each tensor operator is independent of each other. From the transfer learning and collaborative optimization point of view, there are indeed opportunities to exploit task similarities/dependencies to accelerate the convergence speed. For example, the paper the reviewer shared with us exploits structure similarities among tensor operators to predict the expected latency reduction. However, as the reviewer also pointed out, how to model this effect is very challenging, especially in a theoretically grounded way. We would like to discuss these and explore them as future work.
> >
> > We hope our response has addressed most of your concerns. We are glad to continue discussion to address any other questions you may have.

---

> > ### Comment · AnonReviewer1 · 2020-11-21
> > **I raise my score as my concerns are addressed.**
> >
> > Hi authors,
> >
> > Thanks for the extensive rebuttal and added experiments. Most of my concerns are addressed.
> > I agree that we should not discourage the publication of original ideas that are developed independently and concurrently.
> > I think accepting this paper is totally reasonable. The discussion and new techniques are beneficial to the ML+compiler community.
> > My last suggestion is to also mention Ansor as a concurrent work in the main body with a comparison. It is better to summarize their pros and cons in the main body. I think a few sentences are enough, while the experiment figures can be put in the appendix.
> >
> > I raised my score to 6.

---

### Official Review · AnonReviewer4 · 2020-10-29
**Review of DynaTune**

**Rating:** 7
**Confidence:** 2

**Review:**

In this paper, the authors develop DynaTune which achieves faster convergence speed to optimize a DNN model when compared to the state-of-the-art DL compiler, AutoTVM. The key idea is a time-slot-based scheduling method based on UCB-type multi-armed bandit policy. At each time, the scheduler chooses an action to maximize the latency reduction. In practice, A Bayesian belief model via MCMC is used to capture current knowledge of the optimization results to predict future performance, which helps make better decisions and expedites the convergence speed. The idea of using MAB in DL compiler is very interesting. The numerical experiments also demonstrate clear advantage of the proposed DynaTune. My concerns are as follows.

1. In the experiments, it would be more convincing to numerically compare with more advanced DL compiler, e.g., Adams et al. (2019); Chameleon in Ahn et al. (2020). The Chameleon is also a RL based approach for DL compiler.

2. The authors used $C=0.2$ when initial latency of an operator is <1ms, and $C=2$ for all the other cases. Can authors provide more justification on the choice of $C$? Some sensitivity analysis could be helpful.

3. In the end of the first paragraph in Section 5 (line 8 on page 6), the sentence is not finished.

---

> ### Author Response · Authors · 2020-11-19
> **Response to Reviewer #4**
>
> We thank reviewer #4 very much for the detailed summary of our paper and the constructive comments. We provide a detailed response below.
>
> **Q1: In the experiments, it would be more convincing to numerically compare with more advanced DL compiler, e.g., Adams et al. (2019); Chameleon in Ahn et al. (2020). The Chameleon is also a RL based approach for DL compiler.**
>
> A1: We understand the reviewer's concern. To help address the reviewer's concern, we added additional experiments on Chameleon in Appendix C. We choose Chameleon because it is also built on top of AutoTVM and more recent. However, we would like to remark that the dynamic scheduling algorithm in DynaTune is compatible and complementary with existing DL compilation techniques, including Adams et al. (2019) and Chameleon in Ahn et al. (2020), both of which focus on optimizing individual operators.
>
> We use the source code of Chameleon from https://github.com/anony-sub/chameleon. We follow the provided instructions and evaluate Chameleon on the same set of models: ResNet-18 and SqueezeNet on CPU, VGG, and Transformer on GPU. We add Figures 19a--19d in Appendix C of the revised manuscript (uploaded) to show the comparison results. Figure 19c and Figure 19d show that although Chameleon is faster than the baseline AutoTVM on VGG and Transformer on GPU, it is much slower than DynaTune to converge to the lowest latency. Different from our approach, Chameleon uses reinforcement learning to more efficiently explore the code transformation space of a tensor operator, but it still sequentially optimizes one operator at a time. Therefore, although the optimization speed of each operator has been improved, the overall convergence speed is still bounded by the least optimized tensor operators. In contrast, DynaTune focuses on improving the convergence speed of multi-tensor-operators and dynamically allocates time to improve the overall convergence speed.
>
> The original Chameleon paper seems to only evaluate its approach on GPU, but doing model inference on CPU is not uncommon. When running on CPU, a bit surprisingly, we observe that Chameleon is slower than the baseline AutoTVM on ResNet-18 (Figure 19a) and SqueezeNet (Figure 19b). We analyze the performance and find that the RL optimizer in Chameleon adds a non-trivial amount of overhead than the default optimizer (simulated annealing + XGBoost cost model) in AutoTVM on CPU. As a result, although Chameleon reduces the hardware measurement cost and the number of iterations to find the best configuration, its optimization time on CPU is longer than the baseline AutoTVM and is much slower than DynaTune on CPU.
>
> Overall, DynaTune is **1.4–4.7 times faster** than Chameleon to reach the same latency.  Although Dynatune is faster than Chameleon, we want to point out that DynaTune can be combined with Chameleon to achieve better performance, at least on GPU.
>
> **Q2: The authors used C=0.2 when the initial latency of an operator is <1ms, and C=2 for all the other cases. Can authors provide more justification on the choice of C? Some sensitivity analysis could be helpful.**
>
> A2: C > 0 is a parameter that enables to control the exploration/exploitation trade-off. We choose a default value of C = 2 suggested by the theory in [1], which we find to be robust to different range of latencies. When the initial latency is <1ms, we empirically find that C=0.2 leads to increased performance, which we report, although we can't theoretically verify it.
>
> [1] P. Auer, N. Cesa-Bianchi, and P. Fischer, "Finite-time Analysis of the Multiarmed Bandit Problem," Machine learning, vol. 47, no. 2, pp. 235–256, 2002.
>
> **Q3: "In the end of the first paragraph in Section 5 (line 8 on page 6), the sentence is not finished."**
>
> A3: We thank the reviewer for the comment. We have fixed the text in the revision.
>
> We hope that our response has mostly addressed the reviewer’s concerns. We are happy to continue a discussion to address any other questions the reviewer may have.

---

> > ### Comment · AnonReviewer4 · 2020-11-19
> > **My comments are fully addressed**
> >
> > Thanks the authors for the detailed response. My comments are fully addressed. So I increased my score.

---

### Author Response · Authors · 2020-11-24
**Response and revision**

We sincerely thank all reviewers for their positive and constructive comments, time, and effort for improving the paper! Apart from the revision that has already been made to the manuscript, we will incorporate the reviewers' additional suggestions in the final version of the paper. We also very much appreciate all reviewers for sharing unique knowledge and observations, which we believe opens opportunities for our future studies.

---

### Decision · Program_Chairs · 2021-01-07
**Final Decision**

**Decision:**

Accept (Poster)

**Comment:**

This paper applies multi-armed bandits to tuning deep learning code optimization. All reviewers agreed that this is an exploratory paper that opens up a new research area. My main criticism is algorithmic. In particular, the paper applies a 20-year old algorithm to a problem with a small number of arms. It is definitely not as impressive as

https://papers.nips.cc/paper/2018/file/f33ba15effa5c10e873bf3842afb46a6-Paper.pdf

who studied a different (but related) problem. The tuning problem in this paper also seems non-stochastic and contextual, while the authors apply a stochastic non-contextual bandit algorithm.

I shared these concerns with the reviewers, who insisted that the application is important enough to justify the acceptance of the paper. I respect their opinion and therefore suggest an acceptance. I encourage the authors to take my comments into account when revising the paper.